# Psychological Variables Related to Developmental Changes during Adolescence—A Comparison between Alpine and Non-Alpine Sport Participants

**DOI:** 10.3390/ijerph17217879

**Published:** 2020-10-27

**Authors:** Martin Niedermeier, Claudia Kogler, Anika Frühauf, Martin Kopp

**Affiliations:** Department of Sport Science, University of Innsbruck, 6020 Innsbruck, Austria; claudia.kogler@student.uibk.ac.at (C.K.); anika.fruehauf@uibk.ac.at (A.F.); Martin.Kopp@uibk.ac.at (M.K.)

**Keywords:** self-esteem, mental health, high-risk sport, mountain exercise, green exercise

## Abstract

Alpine sport is a popular form of exercise and provides several skills that are potentially relevant for positive development during adolescence. However, empirical data on differences between alpine and non-alpine sport participants in variables related to developmental changes are lacking. Therefore, the primary aim of the present study was to analyze differences in self-esteem and additional variables between adolescent alpine and non-alpine sport participants. A comparison to non-regular exercisers was conducted for self-esteem. In a cross-sectional design, information on self-esteem, sensation seeking, agency, and emotion regulation was collected in 183 adolescents [(mean age: 15.4 (SD: 2.3) years, 71.0% female)]. Alpine sport participants reported significantly higher self-esteem compared to non-regular exercisers, *p* = 0.003, *d* = 0.95, but not compared to non-alpine sport participants, *p* = 0.774, *d* = 0.06. When controlling for sex and high-risk sport engagement, alpine sport participants showed a significantly higher experience of agency compared to non-alpine sport participants, *p* = 0.016, *d* = 0.46. We conclude that alpine sport participation is less relevant with regard to self-esteem compared to regular exercise. However, the characteristics of alpine sport might provide a trigger for higher experience of agency during sport participation, potentially helping to satisfy the increased need for autonomy and independence in adolescence.

## 1. Introduction

Alpine sport, also referred to as mountain exercise, comprises a variety of different forms of exercise, e.g., mountain hiking, mountain biking, climbing, skiing, or snowboarding [1,2]. Alpine sport is increasingly popular in Austria, where three out of five of the most favorite forms of exercise were alpine sports in 2017 (mountain biking, hiking, and skiing) [3]. Although various forms of exercise are summarized in the term alpine sport, the different forms have certain characteristics in common: Firstly, alpine sport is conducted outdoors. Secondly, although exercising in alpine environments is primarily associated with physical and mental health benefits [4,5,6,7], a certain risk for injuries is present when engaging in alpine sport [1,2,8]. Thirdly, the alpine environment is less structured and more complex compared to other sports (e.g., a slope without traces in freeriding vs. a playing field in a ball sport). This combination of characteristics (outdoors, certain risk component, variable environmental conditions) provides a situation where several psychological variables connected to the development of children and adolescents, such as the need for reward, prestige, and risk-taking, might be satisfied in a socially accepted way [9,10].

A variety of psychological variables has been connected to positive developmental changes in adolescence, out of which self-esteem seems to be of high importance. Self-esteem is generally considered as a self-evaluation including personal worth [11] and is regarded as an important trait in the healthy development of children and adolescents, where the phase of adolescence (approximately from age 10 to 20 years) seems to be crucial [12]. Low self-esteem is related to several mental health problems including depression [13], substance dependence, and lower levels of life satisfaction [14]. Therefore, high values of self-esteem are desirable and preventive measures against a decline in self-esteem in adolescents are recommended [12]. Based on meta-analytic evidence, exercise is a promising intervention to improve self-esteem in adolescents [15]. An important aspect in the positive relationship between exercise and self-esteem seems to be regular engagement in exercise [16]. Consequently, non-regular exercisers, i.e., persons who do not engage in exercise on a regular basis, are at risk of low-self-esteem. Potential moderating effects of different modes of exercise seem to be less studied. Therefore, it is unknown if adolescents who engage in exercise *not* specifically conducted in alpine regions, i.e., non-alpine sport participants, show similar self-esteem compared to alpine sport participants. However, there is some evidence that engagement in high-risk sport (traditional rock climbing) is associated with higher self-esteem in an adult population [17] and outdoor exercise in green environments is associated with improved self-esteem [18]. 

In sports containing a certain risk component, at least three further psychological constructs are regularly studied: Sensation seeking is defined by the search for complex and novel experiences combined with an acceptance of certain risks during the search [19]. It is discussed as a correlate of positive risk taking, which is theorized to be associated with beneficial side effects, e.g., on well-being and on adolescent development [20]. In addition to sensation seeking, which has been criticized in recent years as being too simplistic as a sole explanation for participation in those sports [21], emotion regulation and agency emerged as relevant variables. The term emotion regulation is used for all processes applied by individuals to control emotions, i.e., to change the kind, timing, experience, and expression of emotions [22]. Agency is defined as the belief to act and decide independently [23]. Both emotion regulation and agency are relevant in the context of sport participation not only during the activity [23], but also after the activity in a possible transfer effect to everyday life [23]. Furthermore, agency and emotion regulation were positively connected to self-esteem [17] and all three constructs represent important factors in the phase of adolescence. Adolescence represents a sensitive phase of development with a heightened sensitivity for novelty and exploration (i.e., increased sensation seeking), autonomy and independence (i.e., agency), and the development of emotional capacities [9]. Therefore, the assessment of sensation seeking, emotion regulation, and agency seems important when studying variables related to developmental changes in adolescent alpine sport participants. A potentially confounding variable is the engagement in high-risk sport. Although the common definition of high-risk sport (serious injury or death possible during the activity) [24] can be applied to a large amount of different sports, not all alpine sports (e.g., slope skiing) are considered high-risk sports [21]. Engaging in high-risk sport is connected with significantly higher emotion regulation and agency during exercising compared to low-risk sport [17]. Therefore, engagement in high-risk sport should be considered as a covariate. 

Following these considerations, the aim of the present study was to analyze differences in several psychological variables related to developmental changes in adolescence between alpine and non-alpine sport participants. Where possible, an additional comparison to non-regular exercisers was conducted. Concordant with existing literature, we hypothesized that alpine sport participants might show higher values in self-esteem compared to non-regular exercisers. We further hypothesized that alpine sport participants might show different self-esteem, sensation seeking, emotion regulation, and agency values compared to non-alpine sport participants. 

## 2. Materials and Methods 

### 2.1. Design and Sample

Using a cross-sectional study design, a link to a web-based questionnaire was provided to the school children by the teacher of the class in two different secondary schools in Austria. To achieve a higher attendance rate, the children were asked to complete the questionnaire within a regular lesson. Participation was on a voluntary basis. Ethical approval in accordance with the Declaration of Helsinki was provided by the Board for Ethical Questions in Science of the University of Innsbruck (#25/2016). Inclusion criteria were (a) attendance in the selected schools and (b) age between 10 and 20 years. 

### 2.2. Questionnaire

The questionnaire consisted of 52 items asking for demographic data (sex and age), regular exercise behavior, engagement in high-risk sport, self-esteem, sensation seeking, emotion regulation, and agency. Engagement in high-risk sport was assessed by providing examples for typical high-risk sport and asking for the frequency of engaging in these sports. Examples for typical high-risk sport (e.g., off-piste freeriding, rock climbing) were selected in accordance with previous literature [21,23]. Pupils engaging regularly (at least once a week) in at least one of the high-risk sports were categorized as high-risk sport participants. Regular engagement was used to guarantee a certain consistency of engagement in high-risk sport (as opposed to pupils who tried a high-risk sport only once).

The German version of the Rosenberg Self-Esteem Scale [11,25] was used to assess self-esteem and was provided to all participants. The unidimensional 10-item scale is scored from 1 (“strongly disagree”) to 4 (“strongly agree”) and allows one to calculate a sum score ranging from 10 (low self-esteem) to 40 (high self-esteem) after recoding reverse scored items. The German version showed acceptable values for internal consistency and convergent validity (Cronbach’s alpha: 0.81, correlation coefficients to self-efficacy: 0.68–0.72) [26]. The Cronbach’s alpha of the scale in the present sample was 0.67.

Sensation seeking, emotion regulation, and agency was assessed using the German version of the Sensation Seeking, Emotion Regulation, and Agency Scale (G-SEAS) [23,27]. G-SEAS was assessed for the time points “while participating” and “after participating” and was only collected for pupils exercising regularly since it refers to anticipated sport participation. At each time point, 14 items asking for three dimensions are answered on a 7 point Likert scale. The mean scores for the subscale at the two time points “while participating” (experience of sensation, experience of emotion regulation, and experience of agency) and “after participating” (satisfaction of sensation need, transfer of emotion regulation, and transfer of agency) range from 1 (low score) to 7 (high score). Model fit and psychometric values of the scale are acceptable (composite reliabilities: 0.83–0.95). Cronbach’s alpha of the subscales in the present sample was >0.63 for time point while and >0.79 for time point after. As a modification to the original G-SEAS, pupils were asked to state the form of exercise they were referring to when completing the G-SEAS. All pupils stating any form of alpine sport (e.g., skiing, snowboarding, rock climbing) and regularly participating in these sports were defined as alpine sport participants. Alpine sports were selected in accordance with previous mountain exercise literature [7]. All pupils stating non-alpine sports (e.g., soccer, dancing, swimming, gymnastics) and regularly participating in these sports were defined as non-alpine sport participants. All pupils not engaging in exercise regularly (<1 times/week) were referred to as non-regular exercisers.

### 2.3. Statistical Analysis

SPSS Statistics version 26 (IBM, New York, NY, USA) was used for statistical analyses. The main analysis consisted of a series of analyses of variance (ANOVA) and analyses of covariance (ANCOVA) on the primary outcomes self-esteem, experience of sensation/emotion regulation/agency, satisfaction of sensation need, transfer of emotion regulation, and transfer of agency. Values for self-esteem were available for all three groups, resulting in a one-factorial between-subject ANOVA with three categories (alpine sport, non-alpine sport participants, non-regular exercisers). For all G-SEAS variables, a one-factorial between-subject ANOVA with two categories (alpine sport, non-alpine sport participants) was used. Sex (female, male) and engagement in high-risk sport (no, yes) were used as covariates in the ANCOVA according to previous research [17]. We did not use age as a covariate since mean age was similar across groups. Simple contrasts with alpine sport participants as a reference category were calculated. Since the assumption of normal distribution (Shapiro-Wilk) was not met, bias-corrected and accelerated bootstrapped 95% confidence intervals (*95% BCa CI*) based on 1000 samples were calculated for the group difference [28]. Cohen’s *d* was used as an effect size [29]. To check the robustness of the results of the ANOVA, a non-parametric Kruskal-Wallis test was applied for self-esteem and for the G-SEAS variables. For self-esteem, Bonferroni-corrected pairwise comparisons were calculated in cases of a significant finding.

Additional analysis included correlation analysis (Spearman correlation coefficient) between the primary outcomes. All values are displayed as mean (SD) and relative (absolute) frequencies unless otherwise stated. *p*-values < 0.05 were considered to indicate statistical significance (two-tailed).

## 3. Results

### 3.1. Demographic Variables and Engagement in High-Risk Sport

In total, *n* = 183 participants completed the questionnaire with a sex distribution of 71.0% (130) female and 29.0% (53) male (Table 1). Mean age was 15.4 (2.3) years. Out of the total sample, 11.5% (21) were alpine sport participants, 69.4% (127) were non-alpine sport participants, and 19.1% (35) were non-regular exercisers. Alpine sport participants who completed the G-SEAS referred to skiing/snowboarding (81.0%, *n* = 17) or rock climbing (19.0%, *n* = 4).

### 3.2. Differences between Alpine Sport Participants, Non-Alpine Sport Participants, and Non-Regular Exercisers

According to the ANOVA, there was a significant main effect of group on self-esteem, *F*(2,180) = 8.41, *p* < 0.001 (Table 2). Simple contrasts showed that alpine sport participants reported significantly higher self-esteem compared to non-regular exercisers, *95% BCa CI*: 1.3–6.1, *p* = 0.003, *d* = 0.95. The comparison between alpine sport participants and non-alpine sport participants was not significant, *95% BCa CI*: −2.4–1.8, *p* = 0.774, *d* = 0.06. Adding sex and high-risk sport to the analysis in the ANCOVA did not change the interpretation of the results, *F*(2,178) = 8.41, *p* = 0.034. The Kruskal-Wallis test revealed an identical interpretation of the results, i.e., a significant main effect of group on self-esteem, *H*(2) = 19.18, *p* < 0.001. Pairwise comparisons according to the Kruskal-Wallis test showed a significant difference between alpine sport participants and non-regular exercisers, *p* = 0.003, but not between alpine sport participants and non-alpine sport participants, *p* < 1.000.

G-SEAS variables were only compared between alpine sport participants and non-alpine sport participants since data from non-regular exercisers were not available for the G-SEAS. Both experience of sensation and satisfaction of sensation need was rated slightly higher in alpine sport participants and experience and transfer of emotion regulation was rated slightly lower in alpine sport participants. However, both ANOVA and ANCOVA indicated no significant difference in these variables between groups, *p* > 0.348. Experience of agency was not significantly different between groups in the ANOVA, *F*(1, 146) = 3.85, *p* = 0.052. When adding sex and high-risk sport as covariates to the ANCOVA, alpine sport participants showed a significantly higher experience of agency compared to non-alpine sport participants, *95% BCa CI*: 0.1–1.0, *F*(1,144) = 5.94, *p* = 0.016, *d* = 0.46. Transfer of agency after participating was not significantly different between groups, *F*(1,146) = 0.25, *p* = 0.618. Non-parametric methods revealed an identical interpretation of the results, i.e., a non-significant difference in experience of agency, when sex and high-risk sport is not accounted for, *H*(1) = 3.76, *p* = 0.053, and no significant difference in all other variables, *H*(1) < 0.89, *p* > 0.348.

### 3.3. Correlational Analyses

Self-esteem was significantly positively correlated with experience of agency and transfer of emotion regulation and agency after participation (Table 3). No significant correlation with self-esteem was found for experience of sensation, satisfaction of sensation need, and for transfer of emotion regulation. All G-SEAS variables showed significant positive associations, *r_S_* > 0.21, *p* < 0.010.

## 4. Discussion

The primary aim of the present study was to analyze possible differences in psychological variables related to developmental changes in adolescence between alpine and non-alpine sport participants. The main results were that adolescent alpine sport participants showed significantly higher self-esteem compared to non-regular exercisers, but not to non-alpine sport participants. When controlling for sex and engagement in high-risk sport, experience of agency while participating was significantly higher in alpine sport participants compared to non-alpine sport participants.

### 4.1. Analysis of Group Differences

Higher values of self-esteem during the time of adolescence are desirable for many reasons [12,13,14]. The current results suggest that alpine sport participants have higher self-esteem compared to non-regular exercisers; a finding that is in line with previous reports on randomized controlled trials summarized in a systematic review [15]. When comparing the values of the exercise groups (independent of alpine sport participation) to a global analysis across 53 nations, mean values of the present study resemble the global mean value of 30.8 [30]. However, non-regular exercisers were clearly below average (approximately one standard deviation of the global sample), underlining the importance of regular exercise for self-esteem. As a potential mechanism of exercise in relation to self-esteem, it is suggested that engaging in exercise results in an increase in physical fitness, which in turn positively affects self-esteem [31].

Given the characteristics of alpine sports (i.e., conducted outdoors, certain risk component, variable environmental conditions), we hypothesized possible add-on effects of alpine sports on self-esteem compared to non-alpine sports. However, this was not found in the present study. Although the hypothesized difference was seen descriptively, the effect size was very small and far from being significant. To the best of our knowledge, the present study represents a first step in the investigation of alpine sport participation and developmental changes in the age of adolescence. Therefore, the possibilities to draw comparisons to existing literature are limited. Previously conducted research in adults is heterogeneous: In intervention studies, no moderating effect of mode of exercise was found (e.g., aerobic vs. strength training) [31]. In a cross-sectional study, higher self-esteem was found in high-risk sport participants compared to low-risk sport participants, but self-esteem was similar between low-risk sport participants and non-exercise controls in adults [17]. One may conclude that the categorization of different forms of exercise (high-risk vs. low-risk, aerobic vs. strength training) is important. According to the present study, the categorization between alpine and non-alpine sport seems less relevant with regard to the present results on self-esteem. Although several components in alpine sports potentially favor self-esteem [8,18], this does not seem to be enough to create group differences to non-alpine sport participants.

Experience of agency while participating was the only dimension where a significant difference between alpine sport and non-alpine sport participants was found when controlling for sex and high-risk sport. Adolescents have an increased need for autonomy and independence [9]. Agency might satisfy this need and is also positively related to self-esteem [17]. In the present sample, a higher experience of agency was found (opposed to transfer of agency). Although it would be desirable if adolescents could transfer the experience of agency from the sport to difficult situations in everyday life [23], a heightened experience of agency during sport participation might help to satisfy the need for autonomy and independence in adolescents. Similar to transfer of agency, no significant differences between alpine and non-alpine sport participants in emotion regulation or sensation seeking were found. Emotion regulation is a skill that could help adolescents to develop emotional and self-regulatory capacities, which are important in the developmental phase of adolescence [9]. In a qualitative study with adolescent high-risk sport participants, adolescents acknowledged emotional regulatory aspects and autonomous decision making as motives for sport participation [32]. The experience of agency and emotion regulation might be more related to the motives of engagement in (alpine) sport [21,23,33]. Based on the present results, we carefully conclude that the transfer effects of agency and emotion regulation are similar in alpine and non-alpine sport participants, although several aspects should be noted: Firstly, the adolescents who completed the G-SEAS referred mainly to skiing/snowboarding and did not further differentiate between slope skiing/snowboarding and freeriding. However, differences in transfer of agency and emotion regulation were reported between slope skiing/snowboarding and freeriding, with higher transfer of agency and emotion regulation in freeriders [21]. Secondly, the number of alpine sport participants was relatively small, which increases the chance of a random finding. Thirdly, the G-SEAS was validated in an adult population [23,27]. It is unclear if adolescents could understand the items of the G-SEAS well. Therefore, additional research with differentiated categorization of alpine sport is needed.

### 4.2. Correlation Analysis

The results of the correlation analysis between self-esteem and the G-SEAS variables is well in accordance with a previous study [17]. Experience of agency (*r* = 0.16), but not experience of emotion regulation (*r* = 0.04), was significantly positively linked to self-esteem in an adult sample of high-risk and low-risk sport participants and non-exercisers [17]. The authors did not assess transfer of agency or emotion regulation [17], which was positively linked to self-esteem in the present study. Our results confirm the proximity of the constructs agency (experience and transfer) and self-esteem.

### 4.3. Limitations

Some limitations have to be considered when interpreting the results of the present study. Firstly, both the schools and the classes were not randomly selected. This fact—connected with the relatively small number of alpine sport participants—makes it difficult to generalize the present findings. Generalizability of the present results is also hampered by the fact that the present sample predominantly consisted of females. We cannot estimate the influence on the results when a more sex-balanced sample of adolescents is studied. Secondly, it is unclear how well the items of the G-SEAS were understood by adolescents. Although the Cronbach’s alpha values largely indicated acceptable values for internal consistency in the present sample, validation of the (G-) SEAS was done in adult samples only [23,27]. In contrast to the Rosenberg Self-Esteem Scale, which was repeatedly used in the age of adolescence [11,25], this might limit the results of the G-SEAS variables in the younger adolescents of the sample. Thirdly, we summarized the variety of different types of exercise conducted in alpine environments as participation in alpine sports. To detect more subtle group differences, a differentiated categorization of alpine sport might be necessary. Fourthly, although we collected some confounding variables with regard to psychological variables related to developmental changes in adolescence, we cannot assure that other confounding variables were missed, e.g., anthropometric variables, different classes, or health status. Fifthly, all limitations connected to a cross-sectional study (e.g., causal links cannot be drawn) using self-reported assessments have to be mentioned (e.g., non-truthfully answered questions or a potential recall bias). The risk for recall bias should particularly be considered for the G-SEAS since the questionnaire asks for a situation while and after participating in sport. This situation had to be recalled by the adolescents.

However, to the best of our knowledge, the present study can be considered a first approach to investigate a potential association between several psychological variables related to developmental changes in adolescence and alpine sport participation.

## 5. Conclusions

Based on the present results, we conclude that alpine sport participants have higher self-esteem compared to non-regular exercisers. However, contrary to our hypothesis, engagement in alpine sport does not seem to result in an additional group difference compared to non-alpine sport participants in an adolescent age group. Therefore, we conclude that engagement in alpine sport is less relevant with regard to self-esteem compared to other variables (e.g., exercising in general). Consequently, interventions to increase self-esteem in adolescents should aim for general exercise participation irrespective of the mode or environment where exercise is conducted. Experience of agency while participating was higher in alpine sport compared to non-alpine sport participants. This difference might help to satisfy the need for autonomy and independence, which is increased in the age of adolescence and could potentially provide a developmental advantage compared to other forms of sport participation. However, this observation awaits further confirmation and research. Based on the present results, different transfer effects of agency or emotion regulation between alpine and non-alpine sport participants seem unlikely. For future research in this field, we recommend that covariates potentially associated with the outcome variables of the present study are accounted for (e.g., sex, engagement in high-risk sport). With regard to sex, a more balanced sample is recommended for future studies. Furthermore, although prospective studies are highly desirable to draw causal conclusions, we suggest having a more differentiated cross-sectional view on alpine sports, taking into account the variety of different types of exercise conducted in alpine environments, as a next step.

## Figures and Tables

**Table 1 ijerph-17-07879-t001:** Characteristics of the study participants separated by group.

Variables	Alpine Sport Participants		Non-Alpine Sport Participants		Non-Regular Exercisers	
	(*n* = 21)		(*n* = 127)		(*n* = 35)	
	%	(n)	%	(n)	%	(n)
Sex						
Female	85.7%	(18)	66.1%	(84)	80.0%	(28)
Male	14.3%	(3)	33.9%	(43)	20.0%	(7)
High-risk sport						
No	76.2%	(16)	84.3%	(107)	100%	(35)
Yes	23.8%	(5)	15.7%	(20)	0%	(0)
	**Mean**	**(SD)**	**Mean**	**(SD)**	**Mean**	**(SD)**
Age, years	15.4	2.1	15.4	2.3	15.8	2.2

**Table 2 ijerph-17-07879-t002:** Self-esteem, G-SEAS variables separated by group including inferential statistics and effect sizes.

Variables	Alpine Sport Participants		Non-Alpine Sport Participants		Non-Regular Exercisers		*p*-Value ^b^	*p*-Value ^c^	*d* ^d^	*d* ^e^
	Mean	(SD)	Mean	(SD)	Mean	(SD)				
Self-esteem (10: low, 40: high)	30.0	(5.0)	29.7	(4.6)	26.3	(3.0)	**<0.001**	**0.034**	0.06	**0.95**
G-SEAS while participating (1: low, 7: high) ^a^										
Experience of sensation	5.0	(1.0)	4.9	(1.2)			0.646	0.505	0.11	
Experience of emotion regulation	4.6	(1.3)	4.9	(1.4)			0.432	0.668	−0.19	
Experience of agency	5.6	(1.0)	5.1	(1.0)			0.052	**0.016**	**0.46**	
G-SEAS after participating (1: low, 7: high) ^a^										
Satisfaction of sensation need	5.0	(1.0)	4.8	(1.5)			0.553	0.349	0.14	
Transfer of emotion regulation	4.6	(1.2)	4.8	(1.3)			0.586	0.659	−0.13	
Transfer of agency	4.8	(1.3)	4.9	(1.3)			0.618	0.737	−0.12	

G-SEAS: German Sensation Seeking, Emotion Regulation and Agency Scale, SD: standard deviation, d: Cohen’s d, ^a^ unavailable for non-regular exercisers, ^b^ according to ANOVA results, ^c^ according to ANCOVA results (covariates: sex and high-risk sport), ^d^ contrast between alpine sport and non-alpine sport participants, *p* for self-esteem = 0.774, ^e^ contrast between alpine sport participants and non-regular exercisers, *p* for self-esteem = 0.003. Bold values indicate significant differences.

**Table 3 ijerph-17-07879-t003:** Inter-correlations (Spearman correlation coefficients) between self-esteem and G-SEAS variables for alpine and non-alpine sport participants (*n* = 148).

	Variables	1	2	3	4	5	6
**1**	Self-esteem						
	G-SEAS while participating						
**2**	Experience of sensation	0.15					
**3**	Experience of emotion regulation	0.01	**0.43**				
**4**	Experience of agency	**0.35**	**0.21**	**0.24**			
	G-SEAS after participating						
**5**	Satisfaction of sensation need	0.06	**0.62**	**0.44**	**0.34**		
**6**	Transfer of emotion regulation	**0.25**	**0.52**	**0.34**	**0.45**	**0.62**	
**7**	Transfer of agency	**0.24**	**0.53**	**0.32**	**0.52**	**0.53**	**0.79**

G-SEAS: German Sensation Seeking, Emotion Regulation and Agency Scale, bold values indicate significant correlations.

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
