# Peer review of "Psychological Variables Related to Developmental Changes during Adolescence—A Comparison between Alpine and Non-Alpine Sport Participants"

_ijerph, 2020, doi:10.3390/ijerph17217879_

Round 1

Reviewer 1 Report

Review Comments

In their article titled “Psychological Variables Related to Developmental Changes during Adolescence – A Comparison between Alpine and non-Alpine Sport Participants” by Niedermeier et al., the authors have investigated the differences in several psychological variables related to developmental changes in adolescence between alpine and non-alpine sport participants, compared with non-regular exercises, and identified differences in self-esteem, sensation seeking, emotion regulation, and agency values among the groups, using a web-based questionnaire in a cross-sectional study among secondary school students. They found that alpine sports participants had higher self-esteem compared to non-regular exercisers, but not compared to non-alpine sport participants.

The article is very well-written and organized. However, I have the following minor concerns:

  1. It will be better if the authors can briefly clarify in the introduction the differences between the non-alpine and non-regular exercises, and also the relation of non-alpine sport with adolescence psychology in general. Also, what are the sports that were under the non-alpine group, and what were under the alpine group in the study?
  2. The study population was skewed toward female participants. The authors might want to clarify this point and discuss how it might affect generalization of the conclusions made by the study, and if there will be any future direction of the study in line with the sex differences, if noted.
  3. Are the values in Table 3 indicating the r values (i.e. correlation co-efficient values)? It would be better to clarify what the values indicate in that table legend.
  4. The authors could have tested biological variables from blood samples etc. that could have a great impact in understanding the underlying physiological changes in such psychological differences in adolescents.
  5. The authors should mention the body mass index (BMI) of the adolescents, as that will provide a better understanding of the physical conditions of the participants. This is because underweight, obesity, or overweight might have a confounding impact on the results obtained.
  6. Also, describing the family history or personal history of other medical conditions (of course including mental illness history) could have provided a better picture of the conclusions.

Reviewer 2 Report

I enjoyed the manuscript that looks at an interesting hypothesis of the effect of different types of exercise/sport on psychological outcomes in adolescents.  I found it enjoyable and interesting to read.

My review focuses on the statistical analysis and I have a few comments and suggestions the author's may want to consider.

  1. The sample here, while not randomly chosen, was taken from two different classrooms.  Did the author's evaluate whether there were any class to class differences?  If so, the statistical analysis could have used a clustering factor on classroom to ensure that any variability between classrooms was appropriately attributed.  The author's show the demographics by sport participation group, but do not mention whether (most importantly) sport participation was consistent across classrooms.
  2. The authors use ANOVA and ANCOVA models for both the Rosenberg and the G-SEAS.  The G-SEAS is an assessment of specific psychological outcomes measured on a categorical scale and is technically ordinal in nature.  There is a long running debate over whether it is appropriate or not to use ANOVA models on ordinal data. I won't enter into the debate, however at a minimum one should make sure that the outcome meets that assumptions of the ANOVA model (i.e. most relevant here given the sample size is that the data appears to be normally distributed and that the variance of the samples being compared is approximately equal).   Did the authors assess the outcomes with respect to their suitability to the ANOVA/ANCOVA models?  If not, I suggest they do so, and if the data does not support using ANOVA models, methods specific for ordinal outcomes should be used instead.
  3. As the G-SEAS was assessed in the sample students both while participating and after participating, was any evaluation of change done?  Rather than solely looking at the group differences while and after, evaluating whether or not one group changed differently than another may add some interesting details to the analysis.
